# COVID-19 induces a hyperactive phenotype in circulating platelets

**Shane P. Comer** [1,2°], **Sarah Cullivan** [3°], **Paulina B. Szklanna** [1,2°], **Luisa Weiss** [1,2°], **Steven Cullen** [4], **Sarah Kelliher** [5], **Albert Smolenski** [6], **Claire Murphy** [1,7], **Haidar Altaie** [8], **John Curran** [9], **Katherine O'Reilly** [3,6], **Aoife G. Cotter** [6,10,11], **Brian Marsh** [6,12], **Sean Gaine** [3,6], **Patrick Mallon** [6,13], **Brian McCullagh** [3,6], **Niamh Moran** [4], **Fionnuala Ní Áinle** [1,5,6,14]*, **Barry Kevane** [1,5,6]*, **Patricia B. Maguire** [1,2,15]*, **On behalf of the COCOON Study investigators** [¶]

1 Conway SPHERE Research Group, Conway Institute, University College Dublin, Dublin Ireland, 2 School of Biomolecular and Biomedical Science, University College Dublin, Dublin, Ireland, 3 Department of Respiratory Medicine, Mater Misericordiae University Hospital, Dublin, Ireland, 4 School of Pharmacy and Biomolecular Sciences, Royal College of Surgeons in Ireland, Dublin, Ireland, 5 Department of Haematology, Mater Misericordiae University Hospital, Dublin, Ireland, 6 School of Medicine, University College Dublin, Dublin, Ireland, 7 Department of Paediatrics, Royal College of Surgeons in Ireland, Dublin, Ireland, 8 SAS UK Headquarters, Wittington House, Henley Road, Medmenham, Marlow, Buckinghamshire, United Kingdom, 9 SAS Institute Ltd, Dublin, Ireland, 10 UCD Centre for Experimental Pathogen and Host Research, Dublin, Ireland, 11 Department of Infectious Diseases, Mater Misericordiae University Hospital, Dublin, Ireland, 12 Department of Critical Care Medicine, Mater Misericordiae University Hospital, Dublin, Ireland, 13 Department of Infectious Diseases, St Vincent's University Hospital, Dublin, Ireland, 14 Department of Haematology, Rotunda Hospital, Dublin, Ireland, 15 UCD Institute for Discovery, University College Dublin, Dublin, Ireland

☯ These authors contributed equally to this work.
¶ Membership of the COCOON Study is provided in the Acknowledgements.
* fniainle@mater.ie (FNÁ); barrykevane@mater.ie (BK); patricia.maguire@ucd.ie (PBM)

**Data Availability Statement:** All relevant data are within the paper and its Supporting Information files.

## Abstract

Coronavirus Disease 2019 (COVID-19), caused by the novel Severe Acute Respiratory Syndrome Coronavirus 2 (SARS-CoV-2), has affected over 30 million globally to date. Although high rates of venous thromboembolism and evidence of COVID-19-induced endothelial dysfunction have been reported, the precise aetiology of the increased thrombotic risk associated with COVID-19 infection remains to be fully elucidated. Therefore, we assessed clinical platelet parameters and circulating platelet activity in patients with severe and nonsevere COVID-19. An assessment of clinical blood parameters in patients with severe COVID-19 disease (requiring intensive care), patients with nonsevere disease (not requiring intensive care), general medical in-patients without COVID-19, and healthy donors was undertaken. Platelet function and activity were also assessed by secretion and specific marker analysis. We demonstrated that routine clinical blood parameters including increased mean platelet volume (MPV) and decreased platelet:neutrophil ratio are associated with disease severity in COVID-19 upon hospitalisation and intensive care unit (ICU) admission. Strikingly, agonist-induced ADP release was 30- to 90-fold higher in COVID-19 patients compared with hospitalised controls and circulating levels of platelet factor 4 (PF4), soluble P-selectin (sP-selectin), and thrombopoietin (TPO) were also significantly elevated in COVID-19. This study shows that distinct differences exist in routine full blood count and other clinical laboratory parameters between patients with severe and nonsevere COVID-19. Moreover, we have determined all COVID-19 patients possess hyperactive circulating

**Funding:** This research was funded by a COVID-19 Rapid Response grant (20/COV/0157) from Science Foundation Ireland awarded to BK, FNÁ and PBM. The funders had no role in study design, data collection and analysis, decision to publish, or preparation of the manuscript.

**Competing interests:** The authors have declared that no competing interests exist.

**Abbreviations:** ACD-A, acid citrate dextrose A; ARDS, acute respiratory distress syndrome; AU, arbitrary unit; BMI, body mass index; COVID-19, Coronavirus Disease 2019; ELISA, enzyme-linked immunosorbent assay; ICU, intensive care unit; LMWH, low molecular weight heparin; MMUH, Mater Misericordiae University Hospital; MPV, mean platelet volume; NLR, neutrophil-to-lymphocyte ratio; PF4, platelet factor 4; PNR, platelet-to-neutrophil ratio; PPP, platelet-poor plasma; PRP, platelet-rich plasma; RT, room temperature; SARS-CoV-2, Severe Acute Respiratory Syndrome Coronavirus 2; sP-selectin, soluble P-selectin; TPO, thrombopoietin; VTE, venous thromboembolism.

platelets. These data suggest abnormal platelet reactivity may contribute to hypercoagulability in COVID-19 and confirms the role that platelets/clotting has in determining the severity of the disease and the complexity of the recovery path.

## Introduction

Over 30 million people globally have been infected since the outbreak of Severe Acute Respiratory Syndrome Coronavirus 2 (SARS-CoV-2) in December 2019, with over 1 million fatalities reported to date [1]. Coronavirus Disease 2019 (COVID-19) is characterised by a marked pro-inflammatory response with fever, elevated inflammatory markers, and clinical and radiological features of pneumonitis being evident among affected individuals [2]. A complex interplay is known to exist between pro-inflammatory pathway activity and blood coagulation activation; this interplay appears to represent a source of morbidity among SARS-CoV-2–infected patients, particularly among those with severe disease [3]. Unexpectedly high levels of venous thromboembolism (VTE) [4–10] have been reported in this patient group and levels of D-dimer (a marker of increased blood coagulation activation) have been shown to correlate with disease severity and appear to be predictive of mortality [11,12]. COVID-19 patients with cardiovascular risk factors (hypertension, diabetes) are particularly at risk of thrombotic events [13–16]. Moreover, recent postmortem studies have shown evidence of widespread thrombosis in pulmonary vasculature and other organs [17,18].

Antiplatelet drugs (such as aspirin) and anticoagulants (such as low molecular weight heparin (LMWH)) are known to be effective in the prevention and treatment of arterial and venous thrombosis [19–21]. High rates of VTE despite LMWH thromboprophylaxis have been reported in COVID-19, and emerging reports of unusual or severe arterial thrombotic events have been published [5,18,22–24]. As both arterial and venous thrombosis can have devastating consequences, efforts to determine the precise molecular mechanisms underlying the unique hypercoagulable state in COVID-19 must be prioritised in order to optimise preventative and therapeutic strategies [25,26].

Circulating platelets, the activity of which are central to haemostasis and thrombosis, are now understood to fulfil a much broader role in the regulation of a myriad of biological processes including inflammation, wound healing, and angiogenesis [27]. Platelet-mediated immune responses induced by viral infection have previously been reported [28–31], and a recent report has demonstrated altered platelet gene expression profiles and functional responses in patients infected with COVID-19 [32]. Moreover, thrombocytopenia has been noted in response to dengue [33,34] and SARS-CoV infection [35], and among in-patients with COVID-19, thrombocytopenia appears to be associated with increased risk of in-hospital mortality [3,36]. Increased mean platelet volume (MPV), a sensitive indicator of circulating platelet activity and a prognostic marker in thromboinflammation [37–39], has also been reported in association with specific viral infections [40].

In the present study, we aimed to characterise patterns of platelet activity among patients infected with COVID-19 relative to that observed in nonaffected hospitalised patients in order to determine if platelet dysfunction or hyperactivity contributes to COVID-19-induced hypercoagulability.

## Methods

### Patient recruitment, blood collection and platelet preparation

Ethical approval to proceed with this study (1/378/2077) was granted by the Institutional Review Board of the Mater Misericordiae University Hospital (MMUH). Anonymised datasets

describing clinical laboratory parameters among hospitalised patients with severe COVID-19 (requiring critical care support; $n$ = 34), nonsevere COVID-19 (not requiring critical care; $n$ = 20), and non-COVID-19–affected medical inpatients (not requiring critical care; $n$ = 20) were compiled from routine clinical testing results. SARS-CoV-2 infection was confirmed in all cases by RT-PCR analysis of nasopharyngeal swab specimens. Informed consent was sought prior to obtaining additional blood samples for analysis from a subgroup of patients with severe COVID-19, nonsevere COVID-19, and hospitalised patients without COVID-19, with patients matched for age, gender, and body mass index (BMI). All patients were recruited at the MMUH, Dublin, Ireland. A group of healthy control donors were also recruited. Inability to provide informed consent, age younger than 18 years, and exposure to antiplatelet agents were contraindications to study inclusion. In particular, the use of aspirin or other antiplatelet agents in the preceding 2 weeks was an absolute contraindication to inclusion as these agents have direct effects on platelet function and would therefore interfere with the interpretation of results. Thromboembolic events during length of hospitalisation were also considered a contraindication to inclusion in the study. The hospitalised control group consisted of individuals admitted to our hospital with acute medical illness (lower respiratory tract infection, urinary infection, or skin infection) and without history of arterial/venous thrombosis. Patients presenting with a major pro-inflammatory/pro-thrombotic state such as recent major surgery/surgical illness, major traumatic injury, or active cancer were excluded. A Cohort diagram outlining patient recruitment to, and inclusion in, the present study, can be found in S1 Fig.

A volume of 30 ml of blood was collected via venepuncture with a 21-gauge needle into a 10-mL acid citrate dextrose A (ACD-A) vacutainer and centrifuged ($200 \times g$ with no brake for 15 min at room temperature (RT). Platelet-rich plasma (PRP) was isolated and the pH adjusted to 6.5 with ACD-A and supplemented with 1 μM $PGE_1$. Platelets were then isolated from plasma by centrifugation ($600 \times g$ for 10 min at RT) and platelet-poor plasma (PPP) was stored at −80˚C. Isolated platelets from patient groups and healthy controls were washed using a modified Tyrode's buffer (130 mM NaCl, 9 mM NaHCO3, 10 mM Tris-HCl, 10 mM Trisodium citrate, 3 mM KCl, 0.81 mM KH2PO4, 9 mM MgCl2 × 6H2O; pH 7.4 with ACD-A) followed by centrifugation ($600 \times g$ for 10 min at RT). Platelets were again resuspended in modified Tyrode's buffer and incubated at 37˚C.

## Platelet ATP secretion assay

Platelets, isolated as above, for the ATP secretion assay were obtained from patients classified as severe COVID-19 ($n$ = 6), nonsevere COVID-19 ($n$ = 4), and hospitalised control ($n$ = 3), as well as healthy controls ($n$ = 6). ATP secretion was measured using a luminescence-based assay, as previously described [41–44]. Agonists and buffer (5 μl) were dispensed in duplicate into 96-well plates, together with 35 μl of washed platelets and incubated for 3 min at 37˚C, with fast orbital shaking. The doses of platelet agonists used to induce platelet ATP secretion were as follows; thrombin: 0.1 U/ml, 0.5 U/ml; U46619: 1 μM, 6.6 μM. Following incubation, 5 μl of ATP-detecting reagent (chronolume; Labmedics, United Kingdom) was dispensed into each well, and luminescence was measured immediately using a Perkin Elmer 1420 96-well plate reader. Data were expressed as the amount of ATP secreted, in luminescence arbitrary units (AUs), and converted to pmoles ATP released per $10^6$ platelets by comparison with the luminescence recorded from an ATP standard (0.4 mM).

## Enzyme-linked immunosorbent assay (ELISA)

ELISAs for thrombopoietin (DTP00b), P-selectin (DPSE00), and platelet factor 4 (DPF40) were purchased from R&D Systems (Abingdon, UK) and performed according to the

manufacturer's instructions. ELISAs were performed using PPP from patients classified as severe COVID-19 (*n* = 6), nonsevere COVID-19 (*n* = 6), and hospitalised control (*n* = 6), as well as healthy controls (*n* = 6). In brief, undiluted or 1:40 diluted PPP was added in triplicate to wells for thrombopoietin and PF4, respectively. After incubation, the plates were washed and incubated with the respective conjugate. For P-selectin, 1:20 diluted PPP was added simultaneously with the P-selectin conjugate and incubated. Again, plates were washed, and the substrate added. The reaction was stopped with stop solution, and the optical density was determined at 450 nm within 30 min on a Dynex DS2 (Dynex Technologies, Worthing, UK). Protein quantification was performed using a 4 parametric logistic or log-logit regression.

## Statistical analysis

Statistical analysis was performed using R (version 4.0.0) and SAS Viya software (version 3.5). Data were tested for normal distribution using a Shapiro–Wilk test. Normally distributed continuous FBC data were presented as mean ± standard deviation and assessed for statistical significance using a one-way ANOVA. Differences in ATP release and protein expression were determined using a two-tailed *t* test. In both cases, *p*-values were adjusted for multiple comparisons using a Holm–Bonferroni post hoc test. Categorical data were described as percentages and significance was tested using a Fisher exact test. Values below 0.05 were regarded as statistically significant. Data visualisation was performed using SAS Viya.

## Results

### Mean platelet volume (MPV) and platelet-to-neutrophil ratio (PNR) are associated with disease severity in COVID-19

Clinical laboratory data were available for 54 patients with confirmed SARS-CoV-2 infection. Within this group, 34 patients had developed severe disease requiring transfer to the intensive care unit (ICU) for ventilatory support, while the remaining 20 patients did not require critical care support at any stage during the course of their hospital admission. Laboratory parameters relating to these 2 distinct subgroups of patients with COVID-19 (severe COVID-19, *n* = 34; nonsevere COVID-19, *n* = 20) were compared to data from an age and gender-matched cohort of hospitalised medical inpatients without COVID-19 (*n* = 20). Day of admission was defined as the day patients were admitted to the MMUH.

Higher levels of D-dimer were observed on the day of hospital admission among the patients who subsequently developed severe COVID-19 (mean ± SD; 4.95 ± 6.36 mg/L) in comparison ($p$ = 0.0012) to that observed in the nonsevere COVID-19 group (1.01 ± 0.75 mg/L) (Table 1). Furthermore, a significantly higher MPV ($p$ = 0.015) and neutrophil count ($p$ = 0.013) in addition to a significantly lower PNR ($p$ = 0.0067) was noted on the day of admission in the group of patients who subsequently developed a severe disease course relative to that seen in the nonsevere group (Fig 1A). Interestingly, these significant differences remained following transfer to intensive care (Fig 1B). Full clinical characteristics of each study group is provided in S1 Table.

While platelet counts did not differ on the day of admission between patients subsequently designated to have had either a severe or nonsevere disease course (Fig 1A), significantly decreased platelet counts were observed among the severe COVID-19 patients at the time of transfer to the ICU relative to the nonsevere group ($p$ = 0.014, Fig 1B). An increased neutrophil-to-lymphocyte ratio (NLR) in severe cases relative to nonsevere was also detected at the time of hospital admission (another previously reported predictor of disease severity) (S2A Fig) [45–47].

**Table 1. Clinical characteristics of hospitalised control patients without COVID-19 on admission and nonsevere and severe patients with confirmed COVID-19 on admission.** Reference ranges of PT, aPTT, Fibrinogen, and D-Dimer are shown in parentheses. Results are presented as mean ± SD.

| | Hospitalised Controls (n = 20) | Nonsevere COVID-19 (n = 20) | Severe COVID-19 (n = 34) | p-value |
|---|---|---|---|---|
| **Age (years)** | 64.2 ± 17.9 | 69.25 ± 17.7 | 59.4 ± 10.5 | 0.066 |
| **Male, n (%)** | 10 (50%) | 13 (65%) | 21 (62%) | 0.682 |
| **PCT (%)** | 0.262 ± 0.11 | 0.246 ± 0.10 | N/A | 0.626 |
| **P-LCR (%)** | 25.40 ± 9.34 | 23.43 ± 7.57 | N/A | 0.47 |
| **PT (s) (10.4–13.0)** | 13.56 ± 6.92 | 13.26 ± 2.02 | 13.87 ± 5.47 | 0.886 |
| **aPTT (s) (25.0–36.5)** | 28.84 ± 14.17 | 29.22 ± 1.99 | 33.08 ± 8.35 | 0.272 |
| **Fibrinogen (g/L) (1.5–4.0)** | 5.46 ± 1.72 | 4.65 ± 1.64 | 5.58 ± 1.74 | 0.544 |
| **D-Dimer (mg/L) (0.0–0.5)** | 2.53 ± 1.62 | 1.01 ± 0.75 | 4.95 ± 6.36 | **0.027** |

aPTT, activated partial thromboplastin time; COVID-19, Coronavirus Disease 2019; PCT, plateletcrit; PLCR, platelet-large cell ratio; PT, prothrombin time.

## Agonist-induced ADP release is dramatically higher in COVID-19 patients compared with non-COVID-19 hospitalised patients

Recently published data have suggested a crucial mechanistic role for platelets in the pathophysiology of COVID-19 [18,32,48]. We assessed dense granule release following platelet activation with various platelet agonists. ATP secretion was used as a surrogate marker for dense granule secretion, given that the ratio of ADP to ATP in platelet dense granules is a constant 2:3 ratio [49].

Platelets from individuals within each group were isolated and activated with low and high concentrations of thrombin and U46619. Platelets from COVID-19 patients exhibited

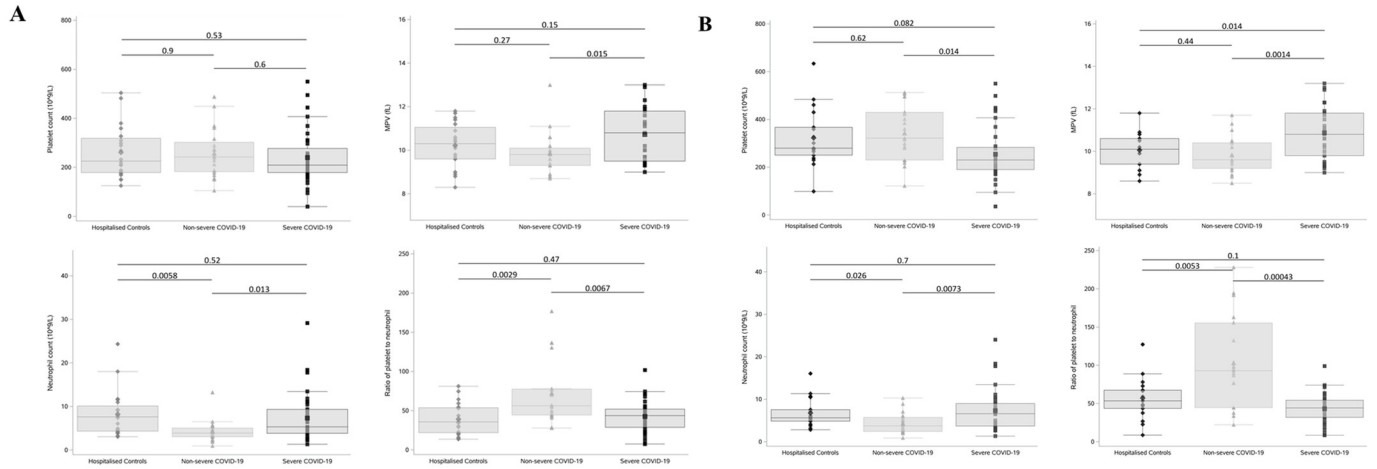

**Fig 1. Routine clinical laboratory parameters are associated with disease severity in COVID-19.** Platelet counts, MPV, Neutrophil counts, and PNRs of patients with severe (n = 34) or nonsevere (n = 20) COVID-19 and hospitalised controls (n = 20). **(A)** On the day of admission, patients who subsequently developed severe COVID-19 had significantly higher MPV (p = 0.015) and neutrophil counts (p = 0.013) and significantly lower PNR (p = 0.0067) compared to those who did not subsequently develop severe COVID-19. **(B)** On day 7 of hospitalisation, nonsevere COVID-19 patients had a significantly higher platelet count compared to severe COVID-19 patients on the day of transfer to intensive care (p = 0.014). Boxplots represent the data median (line inside the box) and the IQR (outline of the box) together with data maximum and data minimum (whiskers) and individual observations (see individual data in S1 Data). Statistical analysis was performed using a two-tailed t test, and p-values were adjusted for multiple comparisons using a Holm–Bonferroni post hoc test. COVID-19, Coronavirus Disease 2019; IQR, interquartile range; MPV, mean platelet volume; PNR, platelet-to-neutrophil ratio.

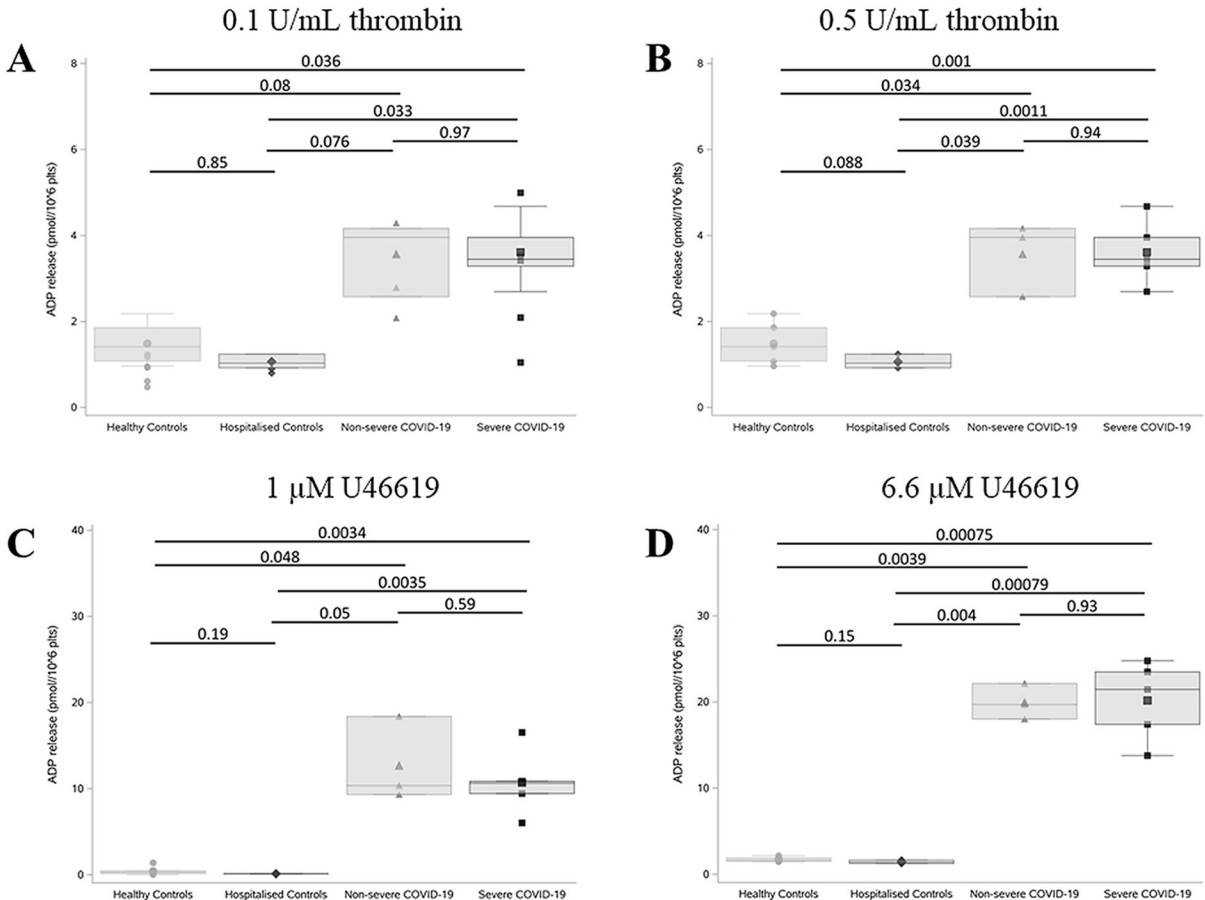

**Fig 2. Agonist-induced ADP release was dramatically higher in COVID-19 patients compared with non-COVID-19 hospitalised patients.** Platelet dense granule release was measured in duplicate in severe ($n = 5$) and nonsevere ($n = 4$) COVID-19 patients compared to hospitalised ($n = 3$) and healthy controls ($n = 6$). Platelets were stimulated with 0.1 U/ml thrombin (**A**), 0.5 U/ml thrombin (**B**), 1 μM U46619 (**C**), and 6.6 μM U46619 (**D**) and ATP release (surrogate for ADP) was measured using a Chronolume luciferase assay. ADP release is expressed as pmol/$10^6$ platelets. Boxplots represent the data median (line inside the box) and the IQR (outline of the box) together with data maximum and data minimum (whiskers) and individual observations (see individual data in S1 Data). Statistical analysis was performed using a two-tailed $t$ test and $p$-values were adjusted for multiple comparisons using a Holm–Bonferroni post hoc test. COVID-19, Coronavirus Disease 2019; IQR, interquartile range.

significantly higher levels of dense granule secretion when compared to both control cohorts (HeC and HoC) with no significant difference between severe and nonsevere COVID-19 patients (Fig 2). Low doses of thrombin (0.1 U/ml) and U46619 (1 μM) induced minimal activation in platelets isolated from healthy and hospitalised control patients. In contrast, dramatically increased dense granule secretion was observed upon platelet activation in patients with COVID-19 (Fig 2A and 2C and S2 Table). Platelet stimulation with 0.1 U/mL thrombin increased ADP-release 3-fold in COVID-19 patients compared with HeC and HoC (Fig 2A and S2 Table). Crucially, platelet activation with the thromboxane A$_2$ receptor agonist U46619 (1 μM) induced a 28-fold increase in ADP-release in COVID-19–positive patients compared to HeC and strikingly, an 89-fold increase between COVID-19–positive patients and HoC (Fig 2C and S2 Table). Platelet stimulation with high-dose thrombin (0.5 U/ml) and U46619 (6.6 μM) induced a 2-fold and 10-fold increase, respectively, in dense granule secretion from COVID-19 patients compared to controls (Fig 2B and 2D and S2 Table). These data suggest that platelets from COVID-19 patients are more responsive to low-dose agonist stimulation

when compared to control platelets, potentially indicating a reduced activation threshold in COVID-19 patients.

## Circulating levels of PF4, sP-selectin, and TPO are significantly elevated in COVID-19

To further characterise this hyperactive platelet phenotype, circulating plasma markers of platelet activity were measured in each of the above study groups ($n = 6$ per group; PPP was available from an additional 2 nonsevere COVID patients and an additional 3 hospitalised controls; clinical characteristics of the study groups are provided in S1 Table). Circulating levels of platelet factor 4 (PF4) were substantially elevated in COVID-19 patients when compared to both hospitalised (HoC) and healthy controls (HeC) ($p = 0.012$ and $p = 0.0033$, respectively; Fig 3A). Nonetheless, circulating PF4 levels could not differentiate between severe and nonsevere COVID-19 patient cohorts ($p = 0.85$). On the other hand, circulating levels of soluble P-selectin (sCD62P) also increased in COVID-19 patients when compared to our control cohorts (Fig 3B) and intriguingly could differentiate between our nonsevere and severe COVID-19 cohorts ($p = 0.037$), suggesting that elevated sP-selectin may be associated with disease severity.

Increased thrombopoietin (TPO) levels have been reported in SARS-CoV infection and more recently in COVID-19 (only in comparison to healthy controls) [50]. Within our study population, TPO levels were found to be significantly higher among patients with COVID-19 (combined severe and nonsevere) compared to HeC ($p = 0.0021$; Fig 3C). No significant difference in circulating TPO levels was noted between COVID-19 and HoC patients ($p = 0.71$; Fig 3C). Interestingly, TPO levels between our severe and nonsevere COVID-19 cohorts showed a clear trend towards elevated circulating TPO in severe COVID-19 infection ($p = 0.052$; Fig 3C).

## Discussion

We have demonstrated that distinct platelet-related patterns in full blood count and other clinical laboratory parameters appear to be associated with the development of a more severe disease course in our cohort of COVID-19 patients, reflecting other recently published data [2,46,51]. Therewithal, we also report novel findings which indicate that COVID-19 is associated with abnormal platelet reactivity. Our observations are likely to be of significant clinical and translational relevance in view of the uncertainty that currently exists with regards to the aetiology of thrombotic risk in COVID-19 and the optimal strategy for prevention and treatment of thrombosis in this population.

In addition to the previously reported observations that neutrophilia, lymphopenia, elevated D-dimer, elevated NLR, and elevated platelet-to-lymphocyte ratio appear to be associated with severe disease, we have reported that within our cohort, a significantly lower PNR was observed on the day of admission among the COVID-19 patients who subsequently progressed to require critical care support [5,11,12,46,52].

Other investigators have demonstrated increased platelet aggregation in response to low-dose agonist stimulation in COVID-19 patients as well as increased platelet thromboxane generation [32]. In line with these findings, we observed increased platelet responsiveness (ADP release) to low-dose agonist stimulation, within our population of COVID-19 patients. Intriguingly, previous reports have demonstrated similar patterns of enhanced platelet reactivity in response to subthreshold concentrations of agonists in the presence of dengue virus nonstructural protein 1 [34,53], potentially indicating a virus-induced sensitisation of platelets. As ADP/ATP and thromboxane A2 are locally acting autocrine agonists which rapidly amplify

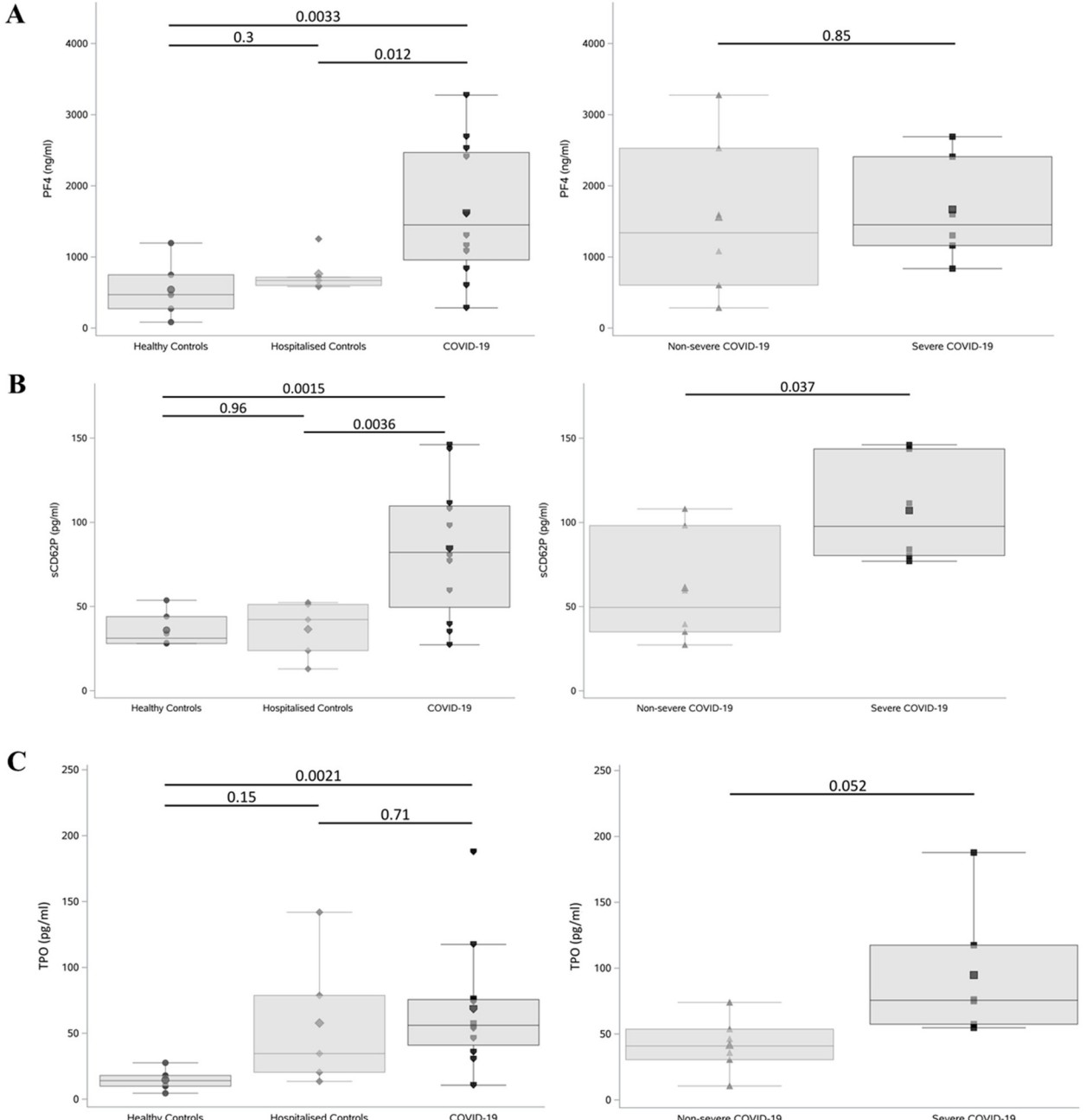

**Fig 3. Circulating levels of PF4, sP-selectin levels, and TPO were significantly elevated in COVID-19.** (**A**) PF4, (**B**) sP-selectin (sCD62P), and (**C**) TPO plasma levels from patients with severe ($n = 6$) and nonsevere ($n = 6$) COVID-19 compared to hospitalised ($n = 6$) and healthy ($n = 6$) controls were measured in triplicate by ELISA. Boxplots represent the data median (line inside the box) and the IQR (outline of the box) together with data maximum and data minimum (whiskers) and individual observations (see individual data in S1 Data). Statistical analysis was performed using a two-tailed *t* test, and *p*-values were adjusted for multiple comparisons using a Holm–Bonferroni post hoc test. COVID-19, Coronavirus Disease 2019; ELISA, enzyme-linked immunosorbent assay; IQR, interquartile range; PF4, platelet-factor 4; sP-selectin, soluble P-selectin; TPO, thrombopoietin.

platelet activation triggering more release of platelet granule contents, our findings provide further evidence that platelets from COVID-19 patients are more reactive, a phenomenon which may contribute to thrombotic risk [4–10].

The circulating levels of the platelet activation markers PF4 and sP-selectin were highly elevated among our cohort of COVID-19–positive patients compared to controls which further supports the hypothesis that COVID-19 is associated with enhanced platelet activation. The elevated levels of TPO in COVID-19 also aligned with previous findings [32]. While circulating levels of PF4 did not differentiate between severe and nonsevere COVID-19 cases, sP-selectin was higher among the severe COVID-19 group, in keeping with previous reports [54]. Other investigators have also reported that sP-selectin levels were elevated in COVID-19 complicated by acute respiratory distress syndrome (ARDS) [55], with even higher levels detected in patients who subsequently died [56]. The mechanism underlying platelet activation in COVID-19 has been speculated to involve MAP kinase pathway [32]; however, the initial trigger remains unidentified. Potential alternative pathways may include internalisation of the virus or the virus genome or binding to other cell surface receptors (e.g., FcγRIIA or dendritic cell-specific intercellular adhesion molecule-3-grabbing nonintegrin), as observed with other viral infections [33,57–60].

This study has a number of limitations. Firstly, our cohort of patients is relatively small, a factor which (at least in part) is likely to be related to the success of the Irish healthcare system in suppressing transmission of the virus during the early stages of the pandemic. Additionally, given the public health restrictions that were in place during the period of this study, recruitment of healthy controls was substantially impeded, and, consequently, the available healthy controls were not appropriately matched for age relative to our COVID-19 group and hospitalised control group. Due to the small sample size, adjustment for confounding factors was not possible.

In conclusion, it has been apparent since the early stages of the COVID-19 pandemic that derangements of haemostasis represent a hallmark of this disease and appear to be associated with significant morbidity [12,61]. Several investigators have reported data suggesting that rates of venous thromboembolism are high in this population and that these thrombotic complications appear to occur despite compliance with standard VTE prevention protocols [5,7]. Reports of unusual and devastating cases of arterial thrombosis are also emerging [18,22–24]. The safety and efficacy of conventional antiplatelet and anticoagulant treatment strategies in the management of this thrombotic risk is becoming a major focus of translational and clinical interest. The aetiology of hypercoagulability in COVID-19 is likely multifactorial, but it is presumed to be driven by the marked inflammatory response which arises following infection [62,63]. It remains to be determined whether standard thrombosis prevention strategies require modification in order to be effective in the prevention of "immunothrombosis" in COVID-19 or if prevention strategies should focus instead on addressing the "upstream" inflammatory response driving coagulation activation. Our data indicate that platelets from patients infected with COVID-19 display a hyperactive phenotype, a factor which may contribute to thrombotic risk. Consequently, further characterisation of platelet dysfunction in COVID-19 and the evaluation of antiplatelet therapy as an adjunct to current thrombosis prevention measures is clearly warranted.

## Supporting information

**S1 Fig. Cohort diagram of patient recruitment and experimental workflow employed for the present study.** nsCOVID-19, nonsevere COVID-19; sCOVID-19, severe COVID-19. (TIF)

**S2 Fig. Additional FBC parameters.** White cell and lymphocyte counts as well as platelet-to-MPV, platelet-to-lymphocyte, platelet-to-white cell, white cell-to-lymphocyte, white cell-to-lymphocyte, and neutrophil-to-lymphocyte ratios on day of hospitalisation **(A)** and 7-day postadmission (hospitalised controls and nonsevere COVID-19) or day of transfer to ICU (severe COVID-19) **(B)** (see individual data in S1 Data). Boxplots represent the data median (line inside the box) and the IQR (outline of the box) together with data maximum and data minimum (whiskers) and individual observations. COVID-19, Coronavirus Disease 2019; FBC, full blood count; ICU, intensive care unit; IQR, interquartile range; MPV, mean platelet volume.
(TIF)

**S1 Table. Clinical characteristics of severe and nonsevere patients with confirmed COVID-19, hospitalised controls without COVID-19, and healthy controls.** Results are presented as mean ± SD.
(DOCX)

**S2 Table. Mean ADP release and fold changes in ADP release upon platelet stimulation with various agonists.**
(DOCX)

**S1 Data. Individual values of data presented in Figs 1–3 and S2 Fig.**
(XLSX)

## Acknowledgments

We thank John Crumlish, Áine Lennon, Jane Culligan, Charlotte Prior, Wendy Keogh and Julie McAndrew of the Department of Pathology, Mater Misericordiae University Hospital, for the help in preparing blood samples and ELISA experiments. We also thank Annette Wallace of Conway Stores for her invaluable help. We also thank all patients who gave consent and donated blood to this study. Finally, we thank all frontline healthcare workers in Ireland for their dedication and professionalism during the COVID-19 pandemic. Additionally, we thank and acknowledge all COCOON Study investigators (all in Ireland unless stated): Peter Doran (Scientific Director, UCD Clinical Research Centre), Jack Lambert (Dept. of Infectious Diseases, Mater Misericordiae University Hospital/UCD School of Medicine), Gerard Sheehan (Dept. of Infectious Diseases, Mater Misericordiae University Hospital/UCD School of Medicine), Tara McGinty (Dept. of Infectious Diseases, Mater Misericordiae University Hospital/UCD School of Medicine), Colman O'Loughlin (Dept. of Critical Care, Mater Misericordiae University Hospital), Jennifer Hastings (Dept. of Critical Care, Mater Misericordiae University Hospital), Mairead Hayes (Dept. of Critical Care, Mater Misericordiae University Hospital), Margaret Doherty (Dept. of Critical Care, Mater Misericordiae University Hospital), Frances Colreavy (Dept. of Critical Care, Mater Misericordiae University Hospital), Ian Conraic-Martin (Dept. of Critical Care, Mater Misericordiae University Hospital), Peter McMahon (Dept. of Radiology, Mater Misericordiae University Hospital/UCD School of Medicine), Leo Lawler (Dept. of Radiology, Mater Misericordiae University Hospital/UCD School of Medicine), Dermot O'Callaghan (Dept. of Respiratory Medicine, Mater Misericordiae University Hospital), Tomas Breslin (Dept. of Emergency Medicine, Mater Misericordiae University Hospital/UCD School of Medicine), Cian McDermott (Dept. of Emergency Medicine; Mater Misericordiae University Hospital/UCD School of Medicine), Robert Turner (Dept. of Critical Care Medicine; Mater Misericordiae University Hospital/UCD School of Medicine), John Crumlish (Laboratory Manager, Mater Misericordiae University Hospital), Aine Lennon (Dept. of

Haematology, Mater Misericordiae University Hospital), Wendy Keogh (Dept. of Haematology, Mater Misericordiae University Hospital), Jane Culligan (Dept. of Haematology, Mater Misericordiae University Hospital), Siobhán McClean (UCD School of Biomolecular & Biomedical Science/UCD Conway Institute), Jennifer Donnelly (Dept. of Obstetrics, Rotunda Hospital), Mary Higgins (Dept. of Obstetrics, National Maternity Hospital), Annemarie O'Neill (Founder, Thrombosis Ireland Patient Organization), Rachel Rosovsky (Massachusetts General Hospital/Harvard Medical School, Massachusetts, USA), Karen Schreiber (Dept. of Rheumatology, Copenhagen University Hospital, Denmark), Cliona Ní Cheallaigh (Dept. of Infectious Diseases/Director of Inclusion Medicine, St James's Hospital/TCD School of Medicine), Mary Cushman (Dept. of Medicine & Pathology, Vermont Medical Center & Larner College of Medicine, University of Vermont, Vermont, USA), Saskia Middeldorp (Academic Medical Centre, Amsterdam, the Netherlands), Peter Juni (Dept. of Medicine, St Michaels Hospital & University of Toronto, Canada), and Michelle Sholzberg (Dept. of Medicine, St Michaels Hospital & University of Toronto, Canada).

## Author Contributions

**Conceptualization:** Shane P. Comer, Sarah Cullivan, Paulina B. Szklanna, Luisa Weiss, Steven Cullen, Albert Smolenski, Claire Murphy, Aoife G. Cotter, Brian Marsh, Sean Gaine, Patrick Mallon, Brian McCullagh, Niamh Moran, Fionnuala Ní Áinle, Barry Kevane, Patricia B. Maguire.

**Data curation:** Shane P. Comer, Paulina B. Szklanna, Luisa Weiss, Haidar Altaie, John Curran, Barry Kevane.

**Formal analysis:** Paulina B. Szklanna, Luisa Weiss, Haidar Altaie, John Curran, Barry Kevane, Patricia B. Maguire.

**Funding acquisition:** Barry Kevane, Patricia B. Maguire.

**Investigation:** Shane P. Comer, Paulina B. Szklanna, Luisa Weiss, Steven Cullen, Fionnuala Ní Áinle, Barry Kevane, Patricia B. Maguire.

**Methodology:** Shane P. Comer, Paulina B. Szklanna, Luisa Weiss, Steven Cullen, Sarah Kelliher, Niamh Moran, Fionnuala Ní Áinle, Barry Kevane, Patricia B. Maguire.

**Project administration:** Shane P. Comer, Paulina B. Szklanna, Luisa Weiss, Fionnuala Ní Áinle, Barry Kevane, Patricia B. Maguire.

**Resources:** Katherine O'Reilly, Barry Kevane.

**Software:** Paulina B. Szklanna, Luisa Weiss.

**Supervision:** Fionnuala Ní Áinle, Barry Kevane, Patricia B. Maguire.

**Visualization:** Paulina B. Szklanna, Haidar Altaie.

**Writing – original draft:** Shane P. Comer, Sarah Cullivan, Paulina B. Szklanna, Luisa Weiss, Steven Cullen, Fionnuala Ní Áinle, Barry Kevane, Patricia B. Maguire.

**Writing – review & editing:** Shane P. Comer, Sarah Cullivan, Paulina B. Szklanna, Luisa Weiss, Steven Cullen, Sarah Kelliher, Albert Smolenski, Claire Murphy, Haidar Altaie, Katherine O'Reilly, Aoife G. Cotter, Brian Marsh, Sean Gaine, Patrick Mallon, Brian McCullagh, Niamh Moran, Fionnuala Ní Áinle, Barry Kevane, Patricia B. Maguire.

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
