## [Editor Report · Decision Letter 0]

11 Aug 2020

Dear Dr Comer, 

Thank you for submitting your manuscript entitled "COVID-19 induces a hyperactive phenotype in circulating platelets" for consideration as a Research Article by PLOS Biology.

Your manuscript has now been evaluated by the PLOS Biology editorial staff [as well as by an academic editor with relevant expertise] and I am writing to let you know that we would like to send your submission out for external peer review.

Please re-submit your manuscript within two working days, i.e. by August 13th, 2020.

Kind regards,

Maya Capelson,

PLOS Biology

---

## [Decision Letter · Decision Letter 1]

16 Sep 2020

Dear Dr. Comer,

Thank you very much for submitting your manuscript "COVID-19 induces a hyperactive phenotype in circulating platelets" for consideration as a Research Article at PLOS Biology. Your manuscript has been evaluated by the PLOS Biology editors, an Academic Editor with relevant expertise, and by several independent reviewers. They all think that you need to improve some aspects before publication.

In particular, both referees and the Academic Editor want to know if the COVID-19 patients were on aspirin. Referee #1 asks about the MPV and platelet activity status in patients with thrombosis and the comparison to those who did not develop clots, considers that the component of the hyperactive platelets needs to be demonstrated (platelet aggregation assays and thrombotic assays), and asks for other factors that were released by the hyperactive platelets and whether a particular factor correlates with disease prognosis. Referee #1 also considers that the authors need to measure P-selectin on the platelet surface instead of soluble, that they need to explain why TPO levels are elevated but there is no thrombocytosis or increased platelets in patients, and proper controls with patients with equal inflammatory states need to be performed. Finally, referee #1 thinks that you need to directly link the hyperactivity of platelets to the hypercoagulable state in COVID-19. Referee #2 asks whether the in-patient control group had a history of thrombotic events, inflammatory diseases/infections, considers that the authors need to comment on fibrinogen levels, and that they need to add details to the methods. In addition, the Academic Editor has kindly suggested to add a flowchart explaining the flow between figures 1, 2, and 3 to help understand why the measures have been performed in a selection of the patients.

In light of the reviews (below), we will not be able to accept the current version of the manuscript, but we would welcome re-submission of a much-revised version that takes into account the reviewers' comments. We cannot make any decision about publication until we have seen the revised manuscript and your response to the reviewers' comments. Your revised manuscript is also likely to be sent for further evaluation by the reviewers.

We expect to receive your revised manuscript within 3 months. 

**IMPORTANT - SUBMITTING YOUR REVISION**

*Re-submission Checklist*

*Published Peer Review*

*PLOS Data Policy*

*Blot and Gel Data Policy*

Sincerely,

Paula

---

Associate Editor,

pjaureguionieva@plos.org,

PLOS Biology

REVIEWS:

Reviewer's Responses to Questions

PLOS authors have the option to publish the peer review history of their article (what does this mean?). If published, this will include your full peer review and any attached files.

Reviewer #1: No

Reviewer #2: No

Reviewer #1: In this manuscript by Comer and colleagues, they present very timely work on the hyperactivity of platelets in COVID-19. This is of interest since there is a high degree of thromboembolic disease in COVID-19 and even with anticoagulation, patients still at risk of thrombosis. In this manuscript the authors demonstrate that patients with COVID-19 have hyperactive circulating platelets and they suggest that this may contribute to the hypercoabulability seen in these patients.

A few concerns that I have include:

1. A large proportion of the patients with more severe presentations of COVID-19 often present with cardiovascular disease such as hypertension or CAD. Since these patients are usually placed on aspirin at baseline and since this manuscript is suggesting the platelet activity is linked to disease severity; it begs the question of whether or not any of these patients were on aspirin. Where there patients who developed covid who were also taking aspirin? Did those patients have hyperactive platelets? Did they have increased thrombotic events?

2. The authors provide information on the mean platelet volume and platelet reactivity in COVID-19 patients. Further analysis of MPV and platelet reactivity as it directly relates to development of a thromboembolic event would directly link hyperactive platelets and hypercoagulability. What was the MPV and platelet activity status in patients with thrombosis and was it different in comparison to those who did not develop clots.

3. Although the platelets were hyperactive by the authors description, a functionality component needs to be demonstrated. Were there changes in platelet aggregation assays? What about thrombotic assays?

4. The authors discuss agonist-induced ADP release. Platelets carry a myriad of inflammatory chemokines and also coagulation factors. What other factors were released from these hyperactive platelets and does a particular factor correlate with disease prognosis?

5. The authors measure soluble P-selectin. P-selectin can come from activated platelets but also activated endothelial cells. We know that COVID-19 is associated with enhanced endothelial injury. The authors should directly measure P-selectin on the platelet surface instead of just soluble P-selectin as a more direct measure of platelet hyperactivity.

6. The authors point out that there is no thrombocytosis in patients. Yet, the TPO levels and MPV are increased which would suggest that the bone marrow is revved up for making platelets. What is keeping the platelet count from becoming elevated? We know that platelets are increased in inflammatory conditions, why is it not increased in COVID which is associated with high levels of inflammation? This especially needs to be explained since the TPO levels are elevated in their study.

7. Although there were differences in the hypereactivity of patients in the nonsevere and severe groups, proper controls with patients with equal inflammatory states need to be performed. Do patients with other highly inflammatory conditions also have platelet hypereactivity? Is this a specific phenomemon on COVID-19.

8. The authors need to directly link the hyperactivity of platelets to the hypercoagulable state in COVID-19

Reviewer #2: This is a well written paper that is highly relevant to the current COVID-19 pandemic. The authors demonstrate differences in routine blood parameters that were associated with disease severity. Platelets form patients with COVID-19 were more prone to activation, demonstrated by ADP release and soluble markers of activation. This study is limited by a small sample size as discuss within the manuscript, nonetheless these are interesting observations that could prove to be clinically useful during the current pandemic.

I have some points to address as detailed below. 

What are the in-patient control group, did any of the patients have a history of thrombotic events, have inflammatory diseases/infections?

The healthy controls were not on aspirin/anti-platelet agents - what about the in-patient group?

Unless I missed it, no reference is made in the text is made to supplementary Table 1

The methods require some more details and some clarification. 

* Please state number of patients recruited per group

* ACD Vacutainer solution A/B?

* Presumably the pH was adjusted with a buffer, eg. modified tyrodes or hepes etc during washing. The methods state it was with ACD?

* What buffer were the isolated platelets resuspended in? In the platelet ATP secretion assay platelet washing is described? If this was not different to the above this information only needs to be stated once

Are data mean +/- IQR? Please state in figure legends.

Can the authors comment on the fibrinogen levels - in other studies these are reportedly raised on admission. 

Typo page 14 - appear to be associated idwith significant morbidity

Comments from the Academic Editor:

It is not clear enough why what measurement was done in which patient. There needs to be more information available on this, and that information is relevant to understanding where the data come from. I am not saying that there need to be more measurement done, but it has to be justified why the measures have been performed in a selection of the patients. A simple flowchart explaining the flow between fig 1, 2, and 3 would already help a lot.

---

## [Decision Letter · Decision Letter 2]

13 Jan 2021

Dear Dr. Comer,

Thank you for submitting your revised Research Article entitled "COVID-19 induces a hyperactive phenotype in circulating platelets" for publication in PLOS Biology. I have now obtained advice from the original reviewers and have discussed their comments with the Academic Editor. 

Based on the reviews, we will probably accept this manuscript for publication, assuming that you will modify the manuscript to address the remaining points raised by the reviewers. Please also make sure to address the data and other policy-related requests noted at the end of this email.

The Academic Editor thinks that the explicit reasons for the selection of patients for the following analyses should be stated explicitly. This is not clear in the Supplementary figure 1 and we suggest that you change this flowchart for a CONSORT style flowchart to give full insight into the selection. Please also complete all the requirements that you will find below my signature in this letter.

We expect to receive your revised manuscript within two weeks.

-  a cover letter that should detail your responses to any editorial requests, if applicable

*Published Peer Review History*

*Early Version*

Sincerely,

Paula

---

Associate Editor,

pjaureguionieva@plos.org,

PLOS Biology

DATA POLICY:

Regardless of the method selected, please ensure that you provide the individual numerical values that underlie the summary data displayed in the following figure panels as they are essential for readers to assess your analysis and to reproduce it: Figures 1A, 1B, 2, 3A, 3B, 3C and Supplementary figures 2A and B.

Please also ensure that **figure legends in your manuscript include information on where the underlying data can be found**, and ensure your supplemental data file/s has a legend.

 ---------------------------

Reviewer remarks:

Reviewer #1: The authors have answered all of my queries and have appropriately revised the manuscript to address concerns raisedd

Reviewer #2: The authors have adequately addressed all my concerns and significantly improve the manuscript.

---

## [Editor Report · Decision Letter 3]

22 Jan 2021

Dear Dr. Comer,

On behalf of my colleagues and the Academic Editor, Bob Siegerink, I am pleased to say that we can in principle offer to publish your Research Article "COVID-19 induces a hyperactive phenotype in circulating platelets" in PLOS Biology, provided you address any remaining formatting and reporting issues. These will be detailed in an email that will follow this letter and that you will usually receive within 2-3 business days, during which time no action is required from you. Please note that we will not be able to formally accept your manuscript and schedule it for publication until you have made the required changes.

PRESS

Thank you again for supporting Open Access publishing. We look forward to publishing your paper in PLOS Biology. 

Sincerely, 

Paula

---

Paula Jauregui, PhD 

Senior Editor 

PLOS Biology
